# Delayed Reconstruction after Major Head and Neck Cancer Resection: An Interdisciplinary Feasibility Study

**DOI:** 10.3390/cancers15102777

**Published:** 2023-05-16

**Authors:** Teresa B. Steinbichler, Tina Rauchenwald, Sasa Rajsic, Hannes T. Fischer, Dolores Wolfram, Annette Runge, Daniel Dejaco, Harald Prossliner, Gerhard Pierer, Herbert Riechelmann

**Affiliations:** 1Department of Otorhinolaryngology, Head and Neck Surgery, Medical University of Innsbruck, 6020 Innsbruck, Austria; 2Department of Plastic, Reconstructive and Aesthetic Surgery, Medical University of Innsbruck, 6020 Innsbruck, Austria; 3Department of Anesthesiology and Critical Care Medicine, Medical University of Innsbruck, 6020 Innsbruck, Austria

**Keywords:** salvage surgery, head and neck cancer, laryngectomy, radiochemotherapy, free flap

## Abstract

**Simple Summary:**

Immediate free flap reconstruction is the method of choice after major head and neck cancer (HNC) resection, but this is frequently associated with long operating hours. This study investigated the feasibility of a two-staged, delayed reconstruction procedure in major HNC.

**Abstract:**

A single immediate reconstruction with free tissue transfer is the method of choice after major head and neck cancer (HNC) resection, but this is frequently associated with long operating hours. Considering regulatory working hour constraints, we investigated whether a two-staged reconstructive approach with temporary defect coverage by an artificial tissue substitute would be feasible. HNC patients underwent either immediate or delayed reconstruction after tumor resection. Patients with delayed reconstruction received preliminary reconstruction with an artificial tissue substitute followed by definitive microvascular reconstruction in a separate, second procedure. Of the 33 HNC patients, 13 received delayed reconstruction and 20 received immediate reconstruction. Total anesthesia time (714 vs. 1011 min; *p* < 0.002) and the total duration of hospital stay (34 ± 13 vs. 25 ± 6 days; *p* = 0.03) were longer in the delayed reconstruction group. Perioperative morbidity (*p* = 0.58), functional outcome (*p* > 0.1) and 5-year postoperative survival rank (*p* = 0.28) were comparable in both groups. Delayed reconstruction after HNC resection was feasible. Perioperative morbidity, functional outcome and overall survival were comparable to immediate reconstruction.

## 1. Introduction

While minor defects following the resection of head and neck cancer (HNC) can be handled with a variety of local flaps, microvascular free tissue transfer has become the standard of reconstruction after major oncologic surgery [1]. Continuous refinements in surgical techniques and technological advances have resulted in high success rates for microvascular free tissue transfer. Immediate, single-staged reconstruction after resection of HNC is the standard approach [2,3]. This common practice is mainly based on a study by Godina on free flap reconstruction after lower extremity trauma. He retrospectively divided patients into three groups, either receiving free tissue transfer within 72 h of injury, between 72 h and 3 months, or 3 months after injury [4]. Less wound infections and superior flap survival rates were observed after immediate reconstruction. However, with improvements in wound care, intensive care management and the introduction of negative pressure wound therapy, this paradigm has been challenged and delayed reconstruction has become the standard of care in lower extremity reconstruction today [5,6].

For HNC, tumor resection followed by microvascular free tissue transfer can lead to long operating times, and hence, surgeon fatigue, longer working hours and working overtime. Often, two surgical teams must be deployed simultaneously, which requires a high logistical effort. In addition, working overtime has been associated with stress and burnout among medical staff. This, in turn, can potentially have a negative impact on patient safety and treatment outcomes [5,6,7]. 

We therefore applied a two-staged procedure with delayed reconstruction in individual HNC patients. In the first procedure, oncologic tumor resection was performed, and the defect was temporarily closed with a synthetic skin substitute. Microvascular reconstruction was performed in a second, separate surgery. This study investigated the feasibility of a two-staged procedure with delayed microvascular reconstruction utilizing temporary wound coverage by an artificial tissue substitute after major head and neck tumor resection.

## 2. Materials and Methods

### 2.1. Study Population

Retrospective data analysis of patients who underwent microvascular free tissue reconstruction after major HNC resection between 2011 and 2021 was performed. A positive vote from the Ethics Committee of the Medical University of Innsbruck had been obtained (UN 1301/2019).

Patients with histologically confirmed mucosal HNC were included. Patients with carcinomas originating from the skin, thyroid gland, eye, brain or spine, and patients with distant tumors metastasizing to the head and neck region were excluded. Treatment was provided as recommended by an interdisciplinary tumor board in accordance with the recommendations of the National Comprehensive Cancer Network (NCCN), as amended [7].

Demographic parameters included age, sex and the American Society of Anesthesiologists Physical Status (ASA) score as a simple measure of general health status [8]. Medical data included the tumor site, the tumor stage according to UICC classification version 7 [9], immunohistochemical p16 status, previous radiotherapy and the phase of the disease (first diagnosis, persistent disease, recurrent disease, second tumor). 

### 2.2. Surgical Procedure

Transoral, transfacial or transcervical approaches were chosen for tumor resections. Total anesthesia time and procedures performed additionally to tumor resection (e.g., tracheotomy, neck dissection) were recorded. Surgical principles followed the NCCN guidelines [7] aiming for histopathologically clear resection margins of at least 5 mm to all sides. Patients received either single-staged, immediate or two-staged, delayed microvascular free tissue reconstruction at the discretion of the surgical teams in charge. In each case, patients were fully informed regarding the procedure and gave informed consent. In patients who underwent delayed reconstruction, the surgical defect was temporarily closed with an artificial skin substitute (EpiGARD^®^, Biovision, Ilmenau, Germany) [10]. The skin substitute also served to cover mucosal defects and was sutured to be as watertight as possible to prevent saliva contamination of the wound area. Postoperatively, all patients were treated with parasympatholytic drugs including scopolamine patches and systemic glycopyrronium bromide. Definitive reconstruction was performed in a second surgery more than 36 h after the tumor resection. Between the two surgeries, patients had to have recovered sufficiently to be transferred back from the intensive care unit (ICU) to the normal inpatient ward.

All patients underwent an enhanced recovery protocol [11] that included standardized perioperative nutritional care, fluid management, thromboembolic prophylaxis [12], antibiotic prophylaxis, preanesthetic medication, urinary catheterization, hypothermia prevention, postoperative nausea prophylaxis, routine postoperative ICU admission, pain management, early postoperative mobilization, postoperative physical pulmonary therapy, tracheostomy care if needed and wound care. Flaps were monitored with hourly postoperative clinical monitoring of flap color, surface temperature, capillary refill and turgor [13], and underwent continuous tissue-oxygen measurement with a Licox Catheter pO2 Micro-Probe instrument for five days [14]. Hemoglobin levels were evaluated immediately after surgery, on day one and two after surgery, and then routinely every third to fourth day in asymptomatic patients. Blood transfusions were administered at hemoglobin levels below 8 g/dL depending on the patients’ symptoms [13,15].

### 2.3. Surgical Outcome Parameters

Outcome parameters included perioperative complications, the total duration of ICU treatment, the total duration of hospital stay and overall survival. 

The severity of perioperative complications after immediate and delayed reconstruction was graded according to the Clavien–Dindo classification [16]. We omitted subclasses ‘a’ and ‘b’ and used an abbreviated five-level classification because of the limited number of patients in this study population. If a patient had more than one complication with different Clavien–Dindo grades, the complication with the highest grade was counted. 

For the survival analysis, the date of tumor resection served as the starting point for calculating survival time.

### 2.4. Functional Outcome Parameters

The Head and Neck Functional Integrity Scale (HNC-FIT scale) served for rapid clinician-rated assessment of functional status during routine oncologic follow-up [17]. The HNC-FIT scale is a matrix of six verbal rating scales reflecting the functional domains of food intake, respiration, speech, pain, mood, and neck and shoulder mobility. Each functional domain is divided into five functional levels. These levels are assessed using verbal ratings, which are anchored to observable external criteria. For example, respiratory function is anchored to the external criterion of the need for a tracheostomy, and food intake is anchored to dependence on a feeding tube. Verbal scores are numerically coded and range from 0 (loss of function) to 4 (normal function). These numeric codes were recorded at each oncologic follow-up visit. Based on a reference population without HNC, patients with scores of 3 and 4 (near normal and normal) were combined as a favorable functional outcome, and patients with scores of 0, 1 and 2 were combined as an unfavorable outcome.

### 2.5. Data Analysis

Frequencies were tabulated and compared with chi-square or Fisher’s exact tests. For metric data, means and standard deviations are provided unless stated otherwise. Ordinal data were compared with the Mann–Whitney U test. Overall survival was analyzed with the Kaplan–Meier method and compared with log rank tests. The median follow-up time was calculated as proposed by Schemper et al. [18]. Statistical analyses were performed using SPSS 27 (IBM Corporation, Armonk, NY, USA).

## 3. Results

### 3.1. Study Population

Between 2011 and 2021, 33 patients (*n* = 33) met the inclusion criteria. Twenty patients (*n* = 20) underwent immediate reconstruction, and thirteen patients (*n* = 13) underwent delayed reconstruction. The mean age of all included patients was 59 (±11) years. Groups were comparable with respect to age (*p* = 0.74), sex (*p* = 0.35), general health according to ASA scores (*p* = 0.20), tumor site (*p* = 0.15), UICC stage (*p* = 0.29) and previous radiotherapy (*p* = 0.6). Data are summarized in Table 1.

### 3.2. Surgical Procedure

In the immediate reconstruction group, resection of the primary tumor was combined with unilateral or bilateral neck dissection in 16/20 (80%) patients. All patients (20/20, 100%) received a temporary or permanent tracheostomy, and 16/20 (80%) patients received a gastrostomy tube. In the delayed reconstruction group, primary tumor resection was combined with unilateral or bilateral neck dissection in all 13 patients (13/13, 100%). A temporary or permanent tracheostomy was performed in 11/13 (85%) patients, and all patients (13/13, 100%) received a gastrostomy tube. The most common type of free flap was a radial forearm flap in both patient groups. Surgical details are listed in Table 2.

Total anesthesia time was 714 (quartiles 619 to 800) minutes in the immediate reconstruction group and 1011 (quartiles 886 to 1126) minutes for both surgical interventions together in the delayed reconstruction group (*p* = 0.002). In the delayed reconstruction group, the median duration of tumor resection was 472 (quartiles 414 to 591) minutes and 473 (quartiles 438 to 531) minutes for reconstruction. The average time interval between the first and second surgery in the staged group was 12.2 (± 7.4, range 5–28) days. A clinical example for delayed reconstruction including temporary defect coverage by an artificial skin substitute (EpiGARD^®^, Biovision, Germany) [10] is shown in Figure 1.

### 3.3. Surgical Outcome

Following immediate reconstruction, 17/20 patients (85%) were transferred to the ICU postoperatively compared to 12/13 patients (92%) following delayed reconstruction, who were transferred to the ICU after tumor resection or reconstruction. The median duration of ICU treatment was 39 (quartiles 20 to 96) hours in the immediate reconstruction group and 36 (quartiles 32 to 108) hours in the delayed reconstruction group (*p* = 0.91). In the delayed reconstruction group, patients spent a median of 16 (quartiles 16 to 18) hours after tumor resection and 20 (quartiles 16 to 94) hours after reconstruction in the ICU.

The median duration of total hospital stay in all 33 patients was 27 (quartiles 22 to 34) days. This was 26 (quartiles 20 to 30) days for patients who had immediate reconstruction and 33 (quartiles 25 to 43) days for patients who had delayed reconstruction (*p* = 0.05). Patients with delayed reconstruction spent a median of 23 (quartiles 17 to 38) days in the hospital after tumor resection and 17 (quartiles 13 to 24) days after reconstruction.

There was no perioperative mortality [19]. The frequency of complications did not differ between the immediate and the delayed reconstruction groups (*p* = 0.58) (Table 3). 

In the immediate reconstruction group, 15/20 (75%) patients experienced complications. Nine patients (9/15, 60%) had one complication, five patients (5/20, 33%) had two complications, and one patient (1/15, 7%) had three complications. The most common complication in the immediate reconstruction group was secondary hemorrhage (7/15, 47%) followed by wound infection (4/15; 27%), which caused the formation of a pharyngocutaneous fistula in one patient (1/15, 7%). In the delayed reconstruction group, nine patients (9/13, 69%) experienced complications. Six patients (6/9, 67%) had one complication, and three patients (3/9, 33%) had two complications. The most common complication in the delayed reconstruction group was also secondary hemorrhage (5/9, 56%) followed by wound infection (3/9, 33%), causing the formation of an orocutaneous fistula in one patient (1/9, 11%). The total count of complications did not differ between groups (*p* = 0.19). The severity of complications according to the Clavien–Dindo classification did not differ between groups (*p* = 0.47) (Table 4). Total flap necrosis occurred in 1/33 (3%) of the cases; the flap necrosis was observed in the immediate reconstruction group (1/20, 5%).

The median follow-up time from the date of tumor resection was 57 (95% CI 23 to 91) months. The mean overall survival was 63 (±8.7) months. Five-year survival in all 33 patients was 52.9%; thus, the median survival could not be calculated. Five-year survival was 55.4% following immediate reconstruction, and 50.8% following delayed reconstruction (*p* = 0.28) (Figure 2).

### 3.4. Functional Outcome

The functional outcome data were available for 21 patients; thirteen patients (13/20, 65%) who had immediate reconstruction and eight patients (8/13, 62%) who had delayed reconstruction. Functional impairment was most often observed for the functional domain of “food intake” in 14 of 21 patients. However, no significant differences between patients after immediate vs. delayed reconstruction were observed in terms of the frequencies of functional impairment for the following functional domains: food intake (*p* = 0.35), breathing (*p* = 0.13), speech (*p* = 0.39), pain (*p* = 0.72), mood (*p* = 0.38), and shoulder and arm mobility (*p* = 0.96). The results are listed in Table 5.

## 4. Discussion

After major resections of head and neck tumors, defect reconstruction with free tissue transfer has become widely accepted. There is solid data demonstrating that functional outcomes are better after microvascular reconstruction than after primary wound healing or other reconstructive techniques. However, major resections of HNC with immediate microvascular reconstruction are time-consuming, and two surgical teams are often required. This implies a high organizational and logistical effort. The long duration of the surgery is a considerable burden for the patient and, in itself, increases the risks of the operation [20].

Tumor resection and microvascular reconstruction took approximately the same amount of time, allowing the long single-staged procedure to be divided into two shorter procedures that were balanced in time. However, the two-staged approach was associated with a significantly longer total anesthesia time (*p* = 0.002) and longer hospital stay (*p* = 0.05). The longer total surgical procedure time with delayed reconstruction results from re-exposure of the surgical site and preparation of the recipient vessels. In a single-staged procedure, recipient vessel preparation is usually performed during neck dissection, saving surgical time. However, in a two-staged procedure, the resection defect and the surgical wound must be closed twice. 

Similarly, the longer overall hospital stay after delayed reconstruction reflects the additional inpatient time between the two surgical procedures. However, the shorter duration of each of the two surgical procedures allows for earlier patient mobilization and a shorter postoperative ICU stay. This may explain the comparable overall ICU time between the one- and two-staged procedures (*p* = 0.91). Between resection and reconstruction, patients could sometimes be discharged home, which was perceived as an alleviation by patients.

The interval between resection and reconstruction also allows evaluation of the resection margins in the definitive histopathology report. This allows focused re-resections during the reconstructive procedure, which are not based on less reliable frozen sections [21]. This occurred in 5 of the 13 patients (38%) who underwent a two-staged procedure, whereby this high number was most likely due to the large size of the resected tumors. In turn, positive resection margins were diagnosed upon definitive histopathology in 3 out of 20 patients (15%) who underwent a one-staged procedure, although intraoperative frozen sections were negative.

One of the essential questions of this study was whether a temporary synthetic skin substitute can provide sufficiently safe defect coverage and protection against salivary contamination when used on mucosal surfaces of the head and neck. The skin substitute material should be non-degradable, waterproof, sufficiently flexible, tear-resistant to allow for a watertight suture and affordable. The skin substitute used in this study (EpiGARD^®^, Biovision, Germany) met these requirements and was used on mucosal sites successfully outside of its approved indication. It is made of a two-layer plastic material, with the top side consisting of a vapor-permeable, but waterproof membrane, and the bottom side consisting of an open matrix of soft polyurethane. It is available in most European countries. In the USA, a similar material, the Integra^®^ Dermal Regeneration Template, is used and approved, e.g., for the temporary reconstruction after skin cancer resection [22,23,24]. Applying a skin substitute for temporary defect coverage, we observed no significant difference in the frequency of wound infections, flap failure or overall complication rate between the immediate and delayed approach. However, we did not perform the two-staged procedure if there was a possibility that large neck vessels would be exposed. Therefore, oral cavity carcinomas, where this is rarely the case, predominated in the delayed reconstruction group.

The previous course of tumor disease was variable in our study population. Surgical resection occurred at the time of initial diagnosis in 18 patients, ≤6 months after the initial diagnosis in 7 patients and >6 months after the initial diagnosis in 8 patients. These time intervals essentially represent upfront surgery, persistent disease and recurrent disease, respectively [25,26]. Interestingly, this had no significant effect on postoperative survival (*p* = 0.64). Univariate analysis of overall survival showed no statistically significant difference in survival between immediate and delayed reconstruction.

Functional outcomes were comparable following immediate and delayed reconstruction. The carefully validated HNC-FIT scales were used to assess functional outcome and do not capture detailed specific functions, but rather higher-level functional domains [17]. There were no clinically relevant differences between immediate and delayed reconstruction in any of the six functional domains studied (Table 4). However, the functional outcome of our study population was significantly worse when compared to an unselected population of over 600 patients with newly diagnosed HNC [27]. This may be explained in part by the fact that complex microvascular reconstruction is used mainly in advanced tumor stages and that this study included patients with recurrent or persistent disease.

This study addressed the question of whether temporary coverage of a large mucosal defect after resection of a HNC with an artificial skin substitute followed by definitive reconstruction with free tissue transfer in a separate surgery is feasible. No data from previous studies in HNC were available in this regard. This explains the small number of patients in this study. No significant disadvantage for the patients in terms of complications, functional outcome and survival (all *p* > 0.2) was observed. However, due to the small study population, the results were subject to a high probability of type II error and do not provide certainty that two-staged procedures are not associated with patient harm. Prospective non-inferiority studies with large case numbers are warranted. 

Additionally, five patients had to be excluded according to the study criteria. They were treated with delayed reconstruction, which was performed within 36 h on the first day after tumor resection. In between the two surgeries, patients spent the night in the ICU. These five patients were comparable to the study population regarding sex, age, ASA score, UICC stage and tumor localization. However, they had a significantly longer total anesthesia time (median 1187, quartiles 1039 to 1309 min; *p* = 0.005), longer ICU treatment (median 314, quartiles 240 to 758 h; *p* = 0.01), longer hospital stay (median 49, quartiles 46 to 74 days; *p* = 0.004), a higher complication rate (5/5, 100%) and Clavien–Dindo score (Grade 1: 1 patient; Grade 2: 2 patients; Grade 4: 2 patients), and a worse overall survival (median survival time 36 months). Thus, we believe that this approach is not feasible. An essential prerequisite for two-staged procedures is the rehabilitation of patients from the ICU back to the normal inpatient ward in order to allow recovery between surgeries for at least one day, preferably longer. Prolonged anesthesia overnight in the ICU followed by reconstruction the next day without an intermediate stay in the normal ward leads to unfavorable survival outcomes.

## 5. Conclusions

Delayed microvascular reconstruction after major head and neck tumor resection with temporal mucosal wound coverage by skin substitutes is feasible. No disadvantages in terms of perioperative morbidity, functional outcome or survival have been observed.

## Figures and Tables

**Figure 1 cancers-15-02777-f001:**
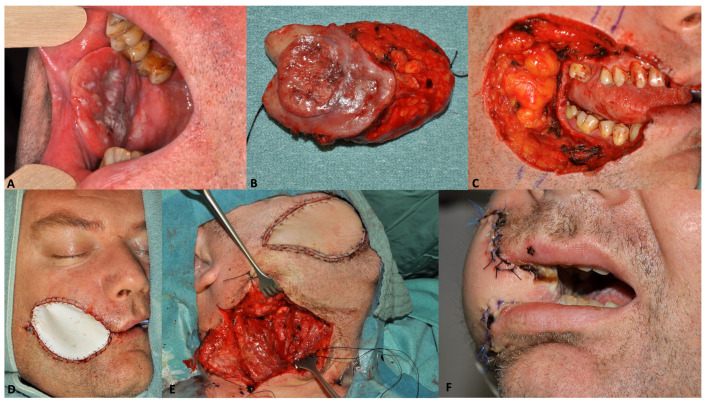
Two-staged, delayed reconstruction in a 44-year-old male patient with oral cavity cancer pT2N2bM0. (**A**) preoperative picture, (**B**) resected tumor, (**C**) intraoperative defect after tumor resection, (**D**) intraoperative site after temporary defect coverage, where the inner and outer linings were temporarily reconstructed with a double layer of EpiGard^®^ ensuring that the mucosa and the outer surface of the wound were both covered with the waterproof side of EpiGard^®^, (**E**) intraoperative defect during second-staged reconstruction with a radial forearm flap, and (**F**) two weeks postoperative.

**Figure 2 cancers-15-02777-f002:**
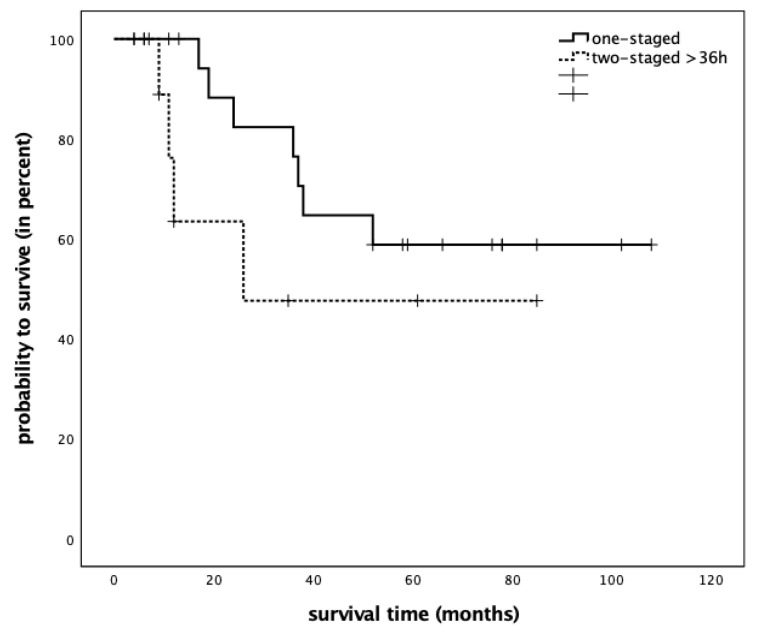
Kaplan–Meier plot comparing overall survival in 33 head and neck cancer patients treated with either one-staged, immediate reconstruction (*n* = 20) or two-staged, delayed reconstruction (*n* = 13) (log rank *p* = 0.28).

**Table 1 cancers-15-02777-t001:** Patient and tumor characteristics. Abbreviations: ASA = American Society of Anesthesiologists, IHC = immunohistochemistry.

Variable	Value	Immediate Reconstruction		Delayed Reconstruction	
		*n* = 20	Percent	*n* = 13	Percent
Sex	Male	15	75.0%	10	76.9%
	Female	5	25.0%	3	23.1%
Age at diagnosis	≤50	4	20.0%	3	23.1%
	51–60	6	30.0%	4	30.8%
	61–70	5	25.0%	3	23.1%
	71–80	5	25.0%	2	15.4%
ASA score	ASA I/II	10	50.0%	7	53.8%
	ASA III/IV	4	20.0%	1	7.7%
	Missing	6	30.0%	5	38.5%
Tumor site	Oral cavity	8	40.0%	12	92.3%
	Pharynx	10	50.0%	1	7.7%
	Esophagus	2	10.0%	0	0.0%
UICC stage	Stage I	1	5.0%	0	0.0%
	Stage II	4	20.0%	0	0.0%
	Stage III	2	10.0%	4	30.8%
	Stage IV	11	55.0%	7	53.8%
	Missing	2	10.0%	2	15.4%
p16 IHC	Negative	13	65.0%	7	53.8%
	Positive	1	5.0%	1	7.7%
	Missing	6	30.0%	5	38.5%
Pack years	<10 pack years	9	45.0%	2	15.4%
	>10 pack years	6	30.0%	6	46.2%
	Missing	5	25.0%	5	38.5%
Prior irradiation	No	13	65.0%	10	76.9%
	Yes	7	35.0%	3	23.1%
Phase of disease	First diagnosis	9	45.0%	8	61.5%
	Persistent disease	1	5.0%	2	15.4%
	Recurrent disease	8	40.0%	3	23.1%
	Second tumor	2	10.0%	0	0.0%

**Table 2 cancers-15-02777-t002:** Surgical characteristics.

		Immediate Reconstruction		Delayed Reconstruction	
Surgical Specification		*n* = 20	Percent	*n* = 13	Percent
Type of free flap	Radial forearm flap	12	60.0%	9	69.2%
	Anterior lateral thigh flap	4	20.0%	3	23.1%
	Latissimus dorsi flap	0	0.0%	1	7.7%
	Jejunal flap	4	20.0%	0	0.0%
Neck dissection	No	4	20.0%	0	0.0%
	Unilateral	8	40.0%	6	46.2%
	Bilateral	8	40.0%	7	53.8%
Tracheotomy	Yes	16	80.0%	11	84.6%
	No	0	0.0%	2	15.4%
	Prior tracheostomy	4	20.0%	0	0.0%
Surgical approach	External approach	7	35.0%	8	61.5%
	Transoral	7	35.0%	4	30.8%
	Total laryngectomy	6	30.0%	1 *	7.7%

* Combined with a hemipharyngectomy in an oropharyngeal cancer patient.

**Table 3 cancers-15-02777-t003:** Type of complications.

	Immediate Reconstruction		Delayed Reconstruction	
Type of Complication	*n* = 20 *	Percent	*n* = 13 *	Percent
Bleeding	7	35.0%	5	38.5%
Wound infection	4	20.0%	3	23.1%
Fistula	1	5.0%	1	7.7%
Pneumonia	2	10.0%	1	7.7%
Thrombosis	1	5.0%	0	0.0%
Other	2	10.0%	1	7.7%

* The overall count is higher than the total number of patients, as individual patients suffered more than one complication.

**Table 4 cancers-15-02777-t004:** Complications by Clavien–Dindo Score.

	Immediate Reconstruction		Delayed Reconstruction	
Clavien–Dindo Score	*n* = 20	Percent	*n* = 13	Percent
0	5	25.0%	4	30.8%
1	4	20.0%	1	7.7%
2	4	20.0%	1	7.7%
3	7	35.0%	6	46.2%
4	0	0.0%	1	7.7%

**Table 5 cancers-15-02777-t005:** Functional outcome using the Aggregated Head and Neck Cancer Functional Integrity (HNC-FIT) scale.

		Immediate Reconstruction	Delayed Reconstruction	
Functional Domain	Functional Integrity	*n* = 13	*n* = 8	*p*-Values
Food intake	Impaired	10 (76.9%)	4 (50.0%)	*p* = 0.35
	Normal/near normal	3 (23.1%)	4 (50.0%)	
Breathing	Impaired	4 (30.8%)	0 (0.0%)	*p* = 0.13
	Normal/near normal	9 (69.2%)	8 (100%)	
Speech	Impaired	8 (61.5%)	3 (37.5%)	*p* = 0.39
	Normal/near normal	5 (38.5%)	5 (62.5%)	
Pain	Impaired	1 (7.7%)	1 (12.5%)	*p* = 0.72
	Normal/near normal	12 (92.3%)	7 (87.5%)	
Mood	Impaired	0 (0.0%)	1 (12.5%)	*p* = 0.38
	Normal/near normal	13 (100%)	7 (87.5%)	
Neck and shoulder mobility	Impaired	5 (38.5%)	3 (37.5%)	*p* = 0.96
	Normal/near normal	8 (61.5%)	5 (62.5%)	

## Data Availability

Data supporting the findings of this study are available from the author upon request.

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
