# Peer review of "Delayed Reconstruction after Major Head and Neck Cancer Resection: An Interdisciplinary Feasibility Study"

_cancers, 2023, doi:10.3390/cancers15102777_

Round 1
Reviewer 1 Report
Thank you very much for giving me the opportunity to review this manuscript. The study is on an interesting topic and provides first information on a surgical approach, which might be interesting for a lot of hospitals struggling with long operating hours in reconstructive surgery in head and neck oncology.
When reading the manuscript I was wondering, on what criteria the decision between a staged or an immediate approach was based on. Did you decide it during the procedure or already beforehand and if so, what were the reasons?
What was the average time interval between first and second surgery in the staged group?
At what time did you perform the neck dissection, during the first or second surgery in the staged group? Can you comment on the preparation on the vessels when you already performed the neck dissection during the first surgery and did you perform the anastomosis at the same side?
Abstract:
When starting to present the results, I would switch the order in the beginning, so better write: “…, 13 received delayed and 20 immediate reconstruction.” – this way it matches with the order you present total anesthesia time and duration of the hospital stay.
Results
Table 2: This table has a lot of mistakes, three times “type of free flap”, two times “Neck dissection”, and so on, always different information?
Table 2: In table 1 there is no case of larynx carcinoma, in table 2 you mention a total laryngectomy – what was the indication here? Or did you perform a laryngopharyngectomy?
Line 161: total duration in immediate reco group 1011 minutes, in the abstract you have a different number, which one is correct?
Figure 1: Please provide complete TNM formula (pT2?N2bcM0).
Sentence starting in line 182: A verb must be missing.
Minor comments:
In the entire manuscript I suggest using “staged” instead of “second-staged” or “two-staged” (maybe also instead of “delayed”), this is more common in the literature.
The quality of the English language is fine, but some sentences could be optimized, sometimes words are simply missing.
Reviewer 2 Report
Dear Authors
This is a very interesting idea. Your paper is well presented. However, I would like to ask some questions:
1. Did you use the Epigard in tongue/FOM/ mandible defects?
2. Did you have delays in getting patients through adjuvant treatment especially the delayed group.
3. How do you explain that the delayed group had slightly worse outcomes although you had the option of re-resection, you operated in less stage 4 cases, no pharyngeal tumours.
4. Please correct table 2
5. What is the cost-benefit repeating a second operation on the same patient. What is the impact on your backlog.
6. Do you have experience with composite free flaps that had delayed reconstruction, please comment.
7. Do you consider the additional surgical stress on you patients? have you noticed changes in the levels of platelets, lymphocytes and albumin for example.
Reviewer 3 Report
The article "Delayed reconstruction with artificial tissue substitute after major head and neck cancer resection: Feasibility and outcomes" presents a novel approach to reconstructing head and neck defects after cancer resection. The study investigates the feasibility and outcomes of a two-staged reconstructive approach using an artificial tissue substitute to cover the defect temporarily, followed by definitive microvascular reconstruction in a separate procedure.
The results of the study indicate that delayed reconstruction using an artificial tissue substitute was feasible and had comparable outcomes to immediate reconstruction. Perioperative morbidity, functional outcome, and overall survival were all similar in both groups. While the delayed reconstruction group did have longer total anesthesia time and hospital stay, this was likely due to the two-stage approach and not the use of the artificial tissue substitute.
This study provides valuable insights into the use of artificial tissue substitutes for temporary coverage of head and neck defects before definitive microvascular reconstruction. By using this approach, the study suggests that operating times can be reduced while still maintaining good outcomes for patients.
However, it should be noted that this is a small-scale study and further research is needed to confirm these findings in larger patient populations. Additionally, careful patient selection and evaluation are critical to ensuring that the use of artificial tissue substitutes is appropriate for each patient and that the benefits of this approach outweigh the risks.
Overall, the article presents a promising approach to reconstructing head and neck defects after cancer resection, and further research in this area is warranted.
